# Lateralised memory networks may explain the use of higher-order visual features in navigating insects

**Giulio Filippi**[1,2]*, **James Knight**[2], **Andrew Philippides**[2], **Paul Graham**[1]

**1** School of Life Sciences, University of Sussex, Brighton, United Kingdom, **2** Department of Informatics, University of Sussex, Brighton, United Kingdom

* gf244@sussex.ac.uk

**Data availability statement:** All code used for this project has been made publicly available on GitHub at https://github.com/giuliofilippi/fpm-paper-code.

## Abstract

Many insects use memories of their visual environment to adaptively drive spatial behaviours. In ants, visual memories are fundamental for navigation, whereby foragers follow long visually guided routes to foraging sites and return to the location of their nest. Whilst we understand the basic visual pathway to the memory centres (Optic Lobes to Mushroom Bodies) involved in the storage of visual information, it is still largely unknown what type of representation of visual scenes underpins view-based navigation in ants. Several experimental studies have suggested ants use "higher-order" visual information – that is features extracted across the whole extent of a visual scene – which raises the question as to how these features might be computed. One such experimental study showed that ants can use the proportion of a shape experienced left of their visual centre to learn and recapitulate a route, a feature referred to as "fractional position of mass" (FPM). In this work, we use a simple model constrained by the known neuroanatomy and information processing properties of the Mushroom Bodies to explore whether the apparent use of the FPM could be a resulting factor of the bilateral organisation of the insect brain, all the whilst assuming a simple "retinotopic" view representation. We demonstrate that such bilaterally organised memory models can implicitly encode the FPM learned during training. We find that balancing the "quality" of the memory match across left and right hemispheres allows a trained model to retrieve the FPM defined direction, even when the model is tested with novel shapes, as demonstrated by ants. The result is shown to be largely independent of model parameter values, therefore suggesting that some aspects of higher-order processing of a visual scene may be emergent from the structure of the neural circuits, rather than computed in discrete processing modules.

**Funding:** J.K., A.P. and P.G. were funded by EPSRC project (EP/S030964/1). A.P. and P.G. were also funded by the BBSRC (BB/X01343X/1). J.K. was additionally funded by EPSRC project (EP/V052241/1). G.F. is funded by the Leverhulme DSP award. The funders had no role in study design, data collection and analysis, decision to publish, or preparation of the manuscript.

## Author summary

Many insects are excellent visual navigators, often relying on visual memories to follow long foraging routes and return safely to their nest location. We have a good understanding of the neural substrates supporting the storage of visual memories in ants. However, it is still largely unknown what type of representation of visual scenes underpins the functions of visual navigation. Experimental studies have suggested ants use "higher-order" features as part of navigation, that is visual features that are extracted across the whole extent of a scene. Using an anatomically constrained model of the insect memory centres, we address the question of whether the use of higher-order visual features may emerge from the overall architecture of the vision-to-memory pathways. We find that balancing the quality of the matches in left and right visual memory provides a potential explanation for some higher-order visual processing and visual cognition suggested by experiments with ants. Overall, this constitutes a contribution to our understanding of the visual cognition and processing of visual scenes used in navigational tasks. We additionally postulate a novel mechanism ants may use to navigate, which is supported by the bilateral structure of the insect brain.

## Introduction

Insects often rely on visual memories to navigate routes within their environment [1–4] and also to relocate goal locations [5–8]. Models of view-based navigation have shown that "snapshot matching" – the comparison of the current view with a single or multiple views committed to memory – can be used for homing [6,9–11] or route following [12–15]. Whilst we have a good understanding of the brain regions supporting visual memories in ants [16–18], much less is known about the visual processing that precedes the memorisation of visual scenes [19]. It is also largely unknown how the bilaterally organised structure of the visual and memory circuits contributes to the functions of visual navigation [20,21]. In this work, we consider behavioural experiments that previously explored several aspects of visual scene perception and visual cognition in ants [22], and explore how the known bilateral neuroanatomy of the visual and memory circuits can help in explaining these experimental results.

Two overarching theories of image processing are the holistic and feature-based approaches, which respectively favour a mostly retinotopic image representation and a parameterized image representation [23]. Whilst much behavioural evidence is consistent with insects learning and using almost "raw" retinotopic images for navigation [19], several experimental studies suggest that insects extract higher-order parameters derived from the whole visual scene. For example, it has been shown that insects can use the centre of mass (CoM) of large shapes [24–27], and theoretical work has proposed that ants could extract rotationally invariant features based on Zernike moments [28,29]. In this work, we focus on the experimental findings of [22], which show that ants can learn a heading direction using the proportion of a shape that lies to the left of their visual centre, a proportion coined 'fractional position of mass' or FPM for short (Fig 1A).

More specifically, in [22], ants were trained to head to a feeder within a cylindrical arena placed at a given location relative to a large shape (visual context). The ants were subsequently tested with different shapes, to see how an imperfect memory match would translate into behavioural decisions. The results of the experiments showed that ants tend to keep the same proportion of a test shape left of their visual centre as they had experienced in training. In

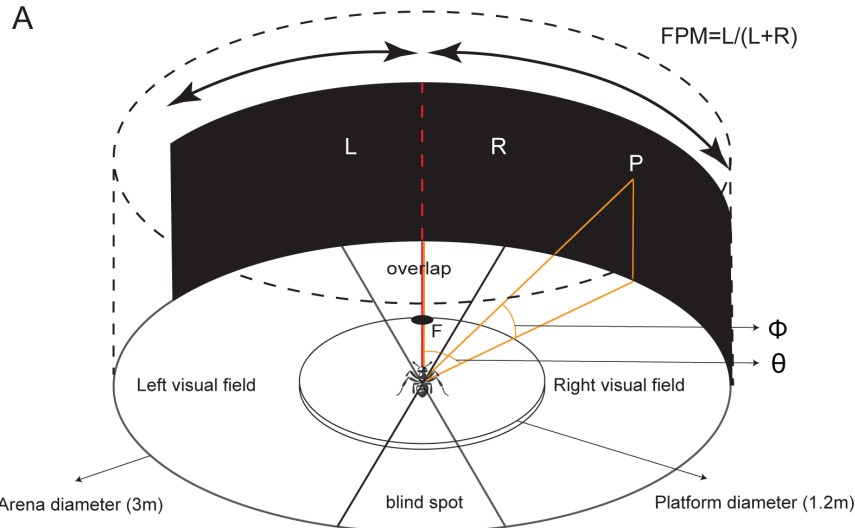

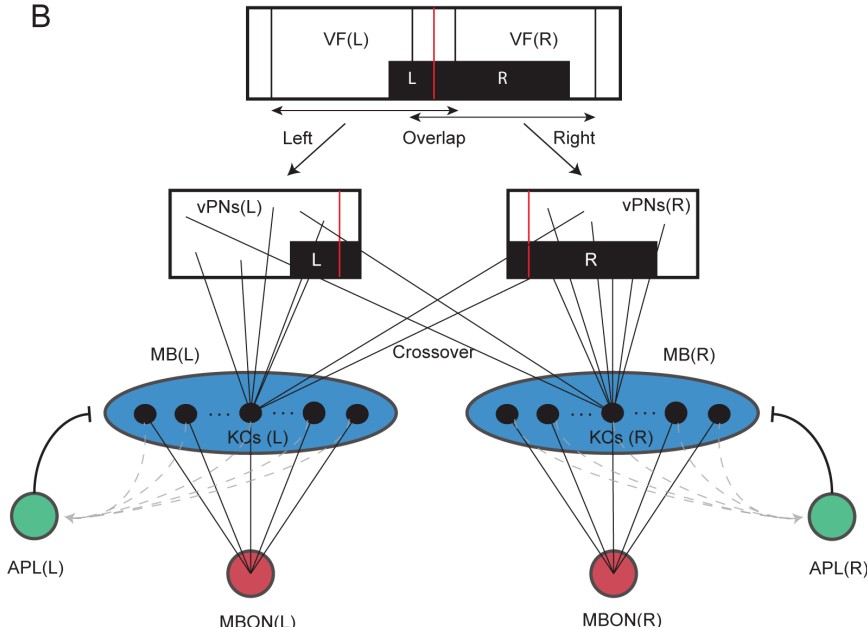

**Fig 1. Experimental setting and network architecture.** (A) Experimental Setting. The experimental setting of [22] consists of a cylindrical arena with 3m diameter and 1.8m high walls. The feeder location is denoted with an F in schematic. The proportion $L/(L + R)$ of a shape occurring on the left of the direct path to the food (centre red line) is referred to as the fractional position of mass (FPM). In the diagram, we show an ant at the centre of the arena (not to scale), as well as its left and right visual fields. In modelling, we assume left and right visual fields have a range of 180 degrees with some of their field of view overlapping in the front (overlap region), leading to a blind spot at the back. $P$ denotes an arbitrary point on the arena wall, and $\theta$ and $\phi$ denote its (azimuthal and polar) stimulus angles as measured from the centre of the arena (orange lines). (B) Network Architecture. We model the ant's memory circuit using an artificial neural network (ANN) with anatomical and information processing features drawn from properties of the insect Mushroom Bodies (MBs). The visual scene in panoramic form is split into left and right visual fields (VFs) by cropping at the appropriate azimuthal angles. The visual projection neurons (vPNs) are taken to be the pixel values of the visual fields. Each of the left and right Mushroom Body compartments contains $N_{KC}$ Kenyon Cells (KCs). vPNs are connected to KCs randomly so that on average KCs have $K$ connections (connections shown only for one KC). The left and right MBs are each recurrently connected to an Anterior Paired Lateral neuron (APLs) which serves to normalise MB activity. Each of the MBs is also fully connected to a Mushroom Body output neuron (MBONs).

other words, ants navigate to the same FPM in test scenarios as the one they are trained with. These findings suggest that ants can extract higher-order features computed over the whole extent of a visual scene to learn and recapitulate routes, in this case the FPM of a large shape. When ants were trained with composite shapes, the experimental results further suggested that ants could segment the visual scene prior to computing the FPM, another example of higher-order visual processing and visual cognition.

Recent lesioning experiments with ants have shown that visual memories are supported by the Mushroom Bodies [17,18], a highly conserved pair of structures in the insect brain known for their role in memory and associative learning [30,31]. Therefore, in this paper, we explore whether navigating wood ants apparent use of the FPM (and other, higher order visual cognitive processes such as shape segmentation) can be explained by a simple model leveraging the known anatomy and information processing properties of these memory centres. We assume a simple "pixel-wise" retinotopic image representation, and propose that, nevertheless, indicators of higher-order visual processing emerge from the functional properties of the downstream memory circuits. Building on recent modelling approaches to visual navigation considering the bilateral organisation of the insect brain [20,21,32–34], our proposed model is equipped with two "eyes" and two Mushroom Bodies (MBs) each with its own output neuron. We allow visual information from the left and right eyes to project into both the left and right MBs, in line with neuroanatomy [35–37]. As a modelling assumption, we introduce a stochastic bias in favour of ipsilateral visual projections so that each MB specialises in processing information from its ipsilateral field of view (lateralisation). We show that, with sufficient bias, a signal for the FPM defined direction is implicitly encoded within the model, and that this result is largely independent of model parameter values. In so doing, we demonstrate that some higher-order processing of the visual scene may be emergent from the bilateral neuroanatomy of the circuits, rather than computed in discrete processing modules.

## Results

### A simple bilateral memory model captures FPM results

In this work, we wanted to explore whether the use of the FPM in navigating ants [22] could result from the bilateral structure of the visual and memory circuits in the insect brain. To do this, we consider a simple agent with an ant inspired visual navigation circuit, within a simulated version of the cylindrical arena from the experiments (Fig 1A). The memory model is implemented as an artificial neural network, retaining the key anatomical and functional properties of the insect Mushroom Bodies (Fig 1B). The visual scene is split into left and right hemispheres prior to projection into the (left and right) Mushroom Bodies (MBs), with a bias favouring ipsilateral connections. Implementational details for each neuron type and the network learning rule are described in detail in the Methods section, and are similar to other approaches taken in the field (e.g.,[38]). For a given training shape, the model is trained on a direct linear path from the centre of the arena to the feeder location, with views facing forward. The trained model is subsequently tested with different shapes, and the informational properties are studied using the outputs of the independent left and right MBONs (which constitute novelty signals) taken across all possible rotations of the agent. An example of left and right rotational novelty signals, alongside their sum and difference are given in (S1 Fig).

In Fig 2, we show the results for the first set of shapes, which consist of large rectangles and trapezoids on a white background. In [22], the distributions of aiming points of the ants are calculated as the distributions of the end points of saccade-like-turns (SLTs) as projected

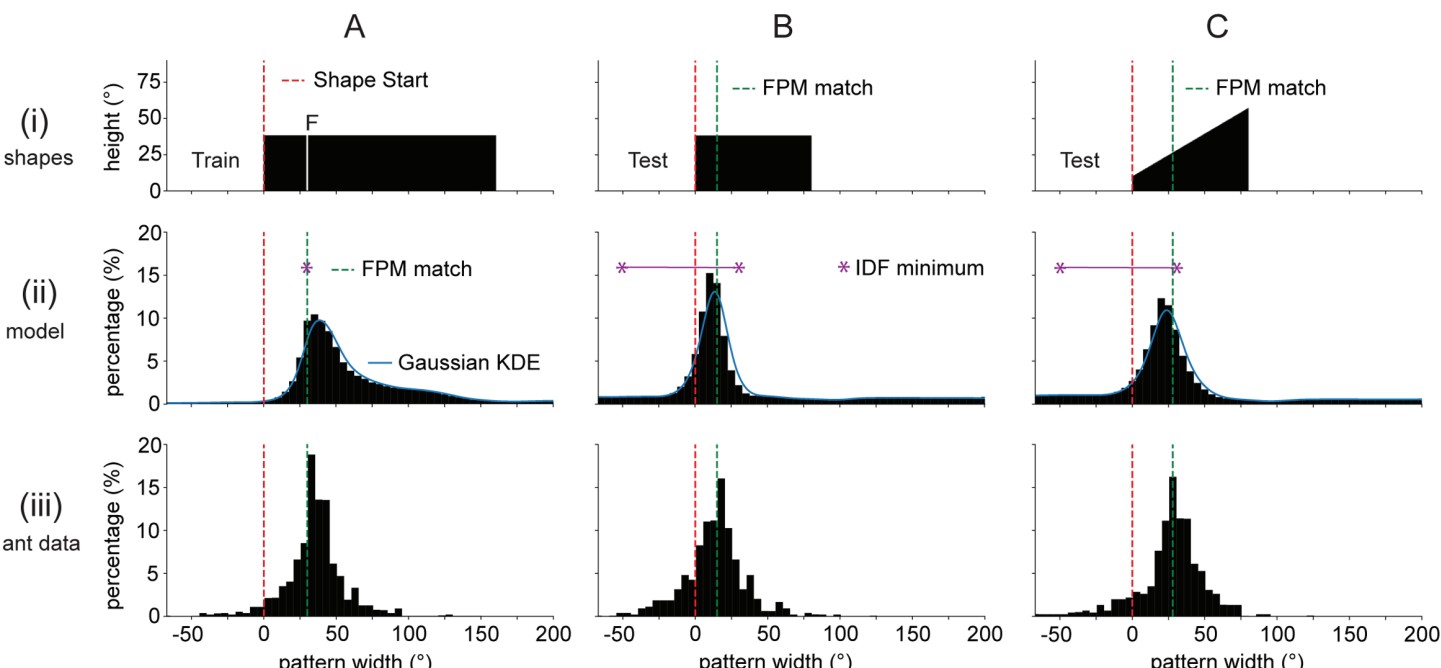

**Fig 2. Results for basic FPM demonstration.** Column (A). (i) The training shape for the first set of experiments consists of a rectangle of width 160° and height 38° (stimulus angles measured from centre of arena). The white line represents the feeder location. (ii) Distribution of model derived headings for the training shape tested against itself. The red dotted line and green dotted line represents the start of the shape and the FPM-match direction respectively. The blue curve is a KDE approximation to the simulated distribution, with gaussian kernel and bandwidth of 5°. The purple asterisk(s) denotes the minimum of the Image Difference Function (IDF), or the angular range over which the IDF is flat. (iii) Experimental goal direction distributions inferred from [22]. Goal directions are measured using the facing direction of saccade-like body turns (SLTs) as projected onto the arena wall [22,39]. (B). Test shape (i), model distribution (ii) and original ant data (iii) for a test shape of a 80° by 38° rectangle. (C). As for (B) but for a trapezoid of width 80°, with heights of 10° and 57° on the left and right sides respectively.

onto the arena wall (row (iii)). SLTs are periods of very high rotational velocity, shown in [39] to be visually guided turns correcting the facing angle towards a goal direction. Therefore, to model the heading distributions, we needed a way to generate a distribution of simulated saccade end points from the outputs of our computational memory model. Whilst it has been shown that the heading directions of ant navigation can often be modelled by minimising a visual novelty signal [12,14,15], our additional modelling insight is that ants, given bilaterally organised memories, might also need to balance left and right novelties (an instance of tropotaxis). Our approach therefore assumes that both the proximity of the novelty sum to its minimum value (low novelty signal), and the proximity of the novelty difference to 0 (lateral balance signal) are important in generating goal directions. To ensure that the observed results are a robust property of our model architecture and not the result of parameter tuning, we report the simulated distributions pooled across a broad range of parameter combinations (see Methods).

In Fig 2 row (ii), we show the simulated distributions for the first series of experiments in [22] alongside the experimentally obtained distributions in row (iii). As can be observed in the plots, the simulated distributions capture the major mode of the experimental findings across all three examples. As was the case in [22], we assume the major mode location in the experimental data encodes the true goal direction of the ants, and therefore we use the distance between simulated modes and true modes as our measure of model performance. Across the three test shapes, the distance between modes between the simulated and

experimental distributions is below 6.1°. If we evaluate the distributions for all parameter combinations and shapes separately, we get an average distance between modes of 5.0° ± 3.5° (mean ± std). The average performance of the model for each of the parameter settings taken independently are shown in (S2A Fig) which demonstrate that the model performs well across all tested parameters. Overall, these results show that the simulated distributions capture the experimentally obtained distributions for simple shapes, and that the results are robust against changes in parameter settings. This suggests that the occurrence of a major mode at the FPM-match direction for this class of shapes is a general principle resulting from the bilateral architecture of the model rather than a specific tuning of its parameter values.

The success of the model in capturing the pattern of results from the biological data is in large part due to the balancing of left and right novelties from the independent MBs. Indeed, a careful analysis shows that there is no signal for the FPM-match direction within the left and right rotational novelty curves taken alone (S1C and S1E Fig), nor is there a consistent signal for the FPM-match direction within the rotational novelty sum (S1D Fig). This finding mirrors the original finding of [22] who reported that Image Difference Function (IDF) models (see e.g., [9]) are inadequate to describe their experimental outcomes. This is because for test shapes smaller than the train shape, the IDF flattens over a large angular interval (in Fig 2, the purple line represents region where IDF is minimal), which leaves the goal direction underdetermined. In our present framework, we can think of the novelty sum as a kind of IDF, which measures the overall match of the train and test shapes, and therefore does not consistently contain a signal for the FPM defined direction. On the other hand, equalising left and right novelties resolves the indeterminacy, and helps in predicting the location of the experimental modes (S1F Fig). In the next section, we study the model in a theoretical setting, and provide an explanation for why we expect left and right novelties to equate at the FPM-match direction for some train and test shape combinations.

## FPM-match direction balances left and right novelties in theoretical analysis for nested shapes

In this section, we provide a theoretical explanation as to why we expect left and right novelties to equate at the FPM-match direction for a variety of train and test shape combinations. To do so we consider a bilaterally organised memory model (as described in the Methods section), with the simplifying assumption that there is a pure lateral split (no crossover connections, and no visual overlap). We also assume that the test shape can be contained (nested) within the train shape, so that we can separate the shape overlap regions into 4 areas: train-left (X), test-left (Y), train-right (W), test-right (Z) (Fig 3A). For instance, this could be the same shape viewed at two different distances, so that one is a scaled down version of the other. The first step is to notice that when both train and test images are aligned at the same FPM value, the following equation holds:

$$\frac{X}{W} = \frac{Y}{Z} \tag{1}$$

If we define $\gamma_L$ and $\gamma_R$ to be the proportion of the train shape covered by the test shape in the left and right visual fields respectively (Fig 3B). Then Equation 1 can be rearranged into:

$$\gamma_L = \frac{Y}{X} = \frac{Z}{W} = \gamma_R \tag{2}$$

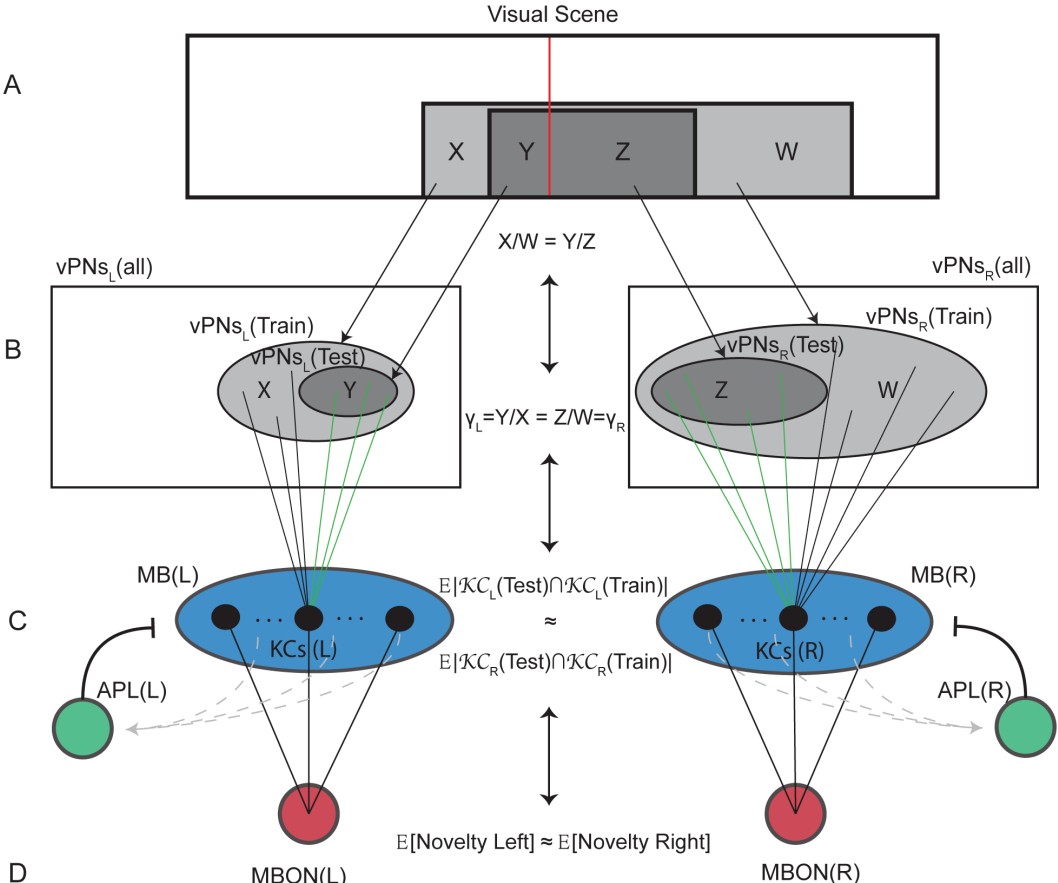

**Fig 3. The theoretical basis for the encoding of FPM by the bilateral memory model.** This figure represents the signal flow as the information progresses through the network layers. (A) Train and test shapes (schematic) are shown in light and dark grey respectively. The panoramic visual scene is split into left and right visual fields (VFs) by a centre line (red line). This splits the two rectangles into two parts which we denote as X and W for the train shape and Y and Z for the test shape. Since we assume the two rectangles are aligned at the same FPM, we have $\frac{X}{W} = \frac{Y}{Z}$. (B) The visual projection neurons split into left and right (vPNs). We show two embedded sets of vPNs for the neurons that would fire for train and test shapes respectively, drawn as ovals in light and dark grey. The equality of FPM values rearranges into an equation that equates laterally segregated informational quantities $\gamma_L = \frac{Y}{X} = \frac{Z}{W} = \gamma_R$. (C) The two Mushrooom Body networks (MBs), each with an anterior paired lateral neuron (APL). We show connections only for one Kenyon Cell (KC) and restrict our focus to those KCs that fired for the train shape, with connections uniformly distributed across the training vPN sets. Connections that are contained ("captured") in the test firing vPN set are shown in green. We also show the expected KC intersection sizes approximately equating across hemispheres. (D) Left and right MBONs which compute the novelty of the bilateral portions of the test shape, relative to the learnt shapes. We show the left and right novelties equating approximately. Equivalence lines running up and down the centre of the image indicate that the chain of causality in the equations can be seen as running up the signal flow as well as down.

The above shows an equivalence between the alignment of the images at the same FPM, and the test shape covering an equal proportion of the train shape in the left and right hemispheres. The second step is to justify why we expect left and right novelties to equate when $\gamma_L = \gamma_R$. To do this, we start with an expression for the expected (Left) test novelty subject to the model having learnt the train image. Given that we assume learning occurs by depreciating the postsynaptic weights of KCs that fire during training (see Methods), we can model the novelty of the test shape as a linear function of the number of KCs that fire on both the train

and test images (Fig 3C and 3D). In equations:

$$\mathbb{E}\big[\text{Novelty Left}\big] = A - B \times \mathbb{E}\big[\big|\mathcal{KC}_{\text{Left}}^{\text{Test}} \cap \mathcal{KC}_{\text{Left}}^{\text{Train}}\big|\big] \tag{3}$$

In the above expression, $A$ and $B$ are positive constants and we use calligraphic notation $\mathcal{KC}$ to denote the sets of KCs (that fire in response to train and test shapes). Using the assumption that KCs are identically distributed, we can rewrite the above expression in terms of the conditional probability of a single Kenyon Cell firing on the test image, given prior knowledge that it fired on the train image:

$$\mathbb{E}\big[\text{Novelty Left}\big] = A - B \times pN_{KC} \times \mathbb{P}\big(\text{KC}_{\text{Left}}^{\text{Test}} = 1 \mid \text{KC}_{\text{Left}}^{\text{Train}} = 1\big) \tag{4}$$

Here $p$ denotes the overall probability of a Kenyon Cell firing, and $N_{KC}$ denotes the number of KCs in one hemispheric MB. From the above formula, we deduce that left and right novelties equate when the conditional firing probabilities equate on the left and right sides. The last step is therefore to justify why the left and right conditional firing probabilities of a single cell $\mathbb{P}\big(\text{KC}_{\text{L/R}}^{\text{Test}} = 1 \mid \text{KC}_{\text{L/R}}^{\text{Train}} = 1\big)$ equate approximately when $\gamma_L = \gamma_R$.

For a Kenyon Cell to fire, it needs to have an appropriate number of coincident inputs from firing vPNs above the population average (e.g., exceeding $(1-p)^{th}$ quantile for our APL model). Given prior knowledge that a KC fired in response to the train image, we know the density of connections of that KC in the train firing vPN set must be significantly over average. The KC therefore also fires in response to the test image if sufficient of those train connections also coincidentally connect within the test firing vPN set (displayed as green connections in Fig 3C). Because the pattern of connections is assumed to be random, the connections to vPNs that fired at train time are randomly distributed homogeneously across the train firing vPN set. As a consequence of this, each connection with a firing vPN is contained ("captured") within the test firing vPN set with a probability given by the covering proportion $\gamma$ ($\gamma_L$ in the left hemisphere and $\gamma_R$ in the right hemisphere). Capturing the same proportion of connections on both sides in turn ensures that conditional distributions get the same "edge" over population distributions in both hemispheres, which explains why $\gamma_L = \gamma_R$ approximately balances left and right conditional firing probabilities and therefore image novelties (Fig 3D; a more detailed explanation is included in S1 Appendix).

In light of the theoretical justification as to why left and right novelties balance (approximately) at the FPM-match direction, we do not expect the result to be sensitive to certain parameter values. Indeed, running the computational model for many simple shapes and parameter settings shows that in almost all cases the rotational novelty difference has a zero-crossing at the FPM-match direction (S3 Fig). This includes weight initialisation decisions, for instance we can initialise all vPN to KC weights randomly, all KC to MBON weights randomly, or all network weights randomly, and the rotational novelty difference still has a zero-crossing at the FPM-match direction in all three cases (S3 Fig row (vi)). The parameters that are the most likely to shift the location of zero-crossings are crossover and overlap parameters. Crossover values close to 0.5 lead to unstable outputs, as the bilateral difference signal becomes weak before flattening at 0.5 (S3 Fig row (iv)). High overlap values can considerably shift the location of zero-crossings, especially for smaller test shapes (S3 Fig row (v)). These observations explain why we decided to pool model predictions over different combinations of the crossover and overlap parameters to better understand the aggregate behaviour of the model (see Methods).

## Modelling assumptions also explain experimental results for more complex shapes

In the previous two sections, we analysed cases where test shapes could be mostly contained within the train shapes. Following the second set of experiments in [22], we consider more complicated interactions between train and test shapes composed of triangles and trapezoids. Whilst the theoretical argument does not directly apply to these train and test shape combinations, because test shapes cannot always be contained within the train shapes, we will show that the same methodology for simulating goal direction distributions can still provide a strong heuristic for the location of modes in the experimental heading data.

The train shape used in this set of experiments is a 160° wide scalene triangle (Fig 4A). The first test shape (B) consists of the train shape reflected horizontally and the second test shape (C) consists of a 160° wide trapezoid. As can be observed in Fig 4 row (ii), the simulated distributions capture the major mode locations of the experimental distributions displayed in row (iii). Across the three examples, the simulated distributions have a distance between modes below 8.7°. If we instead split the analysis over all parameter values and shapes, the average distance between modes is 7.3° ± 5.7° (mean ± std). The average performance of the model for each of the parameter settings considered is shown in (S2B Fig). These results show that the model still robustly estimates experimental mode locations.

In this set of experiments, two different colonies were reported to have different behaviours on the second test shape, as displayed in the two diverging distributions (black and blue) of Fig 4B(iii). The distribution of heading directions for the second colony (Colony

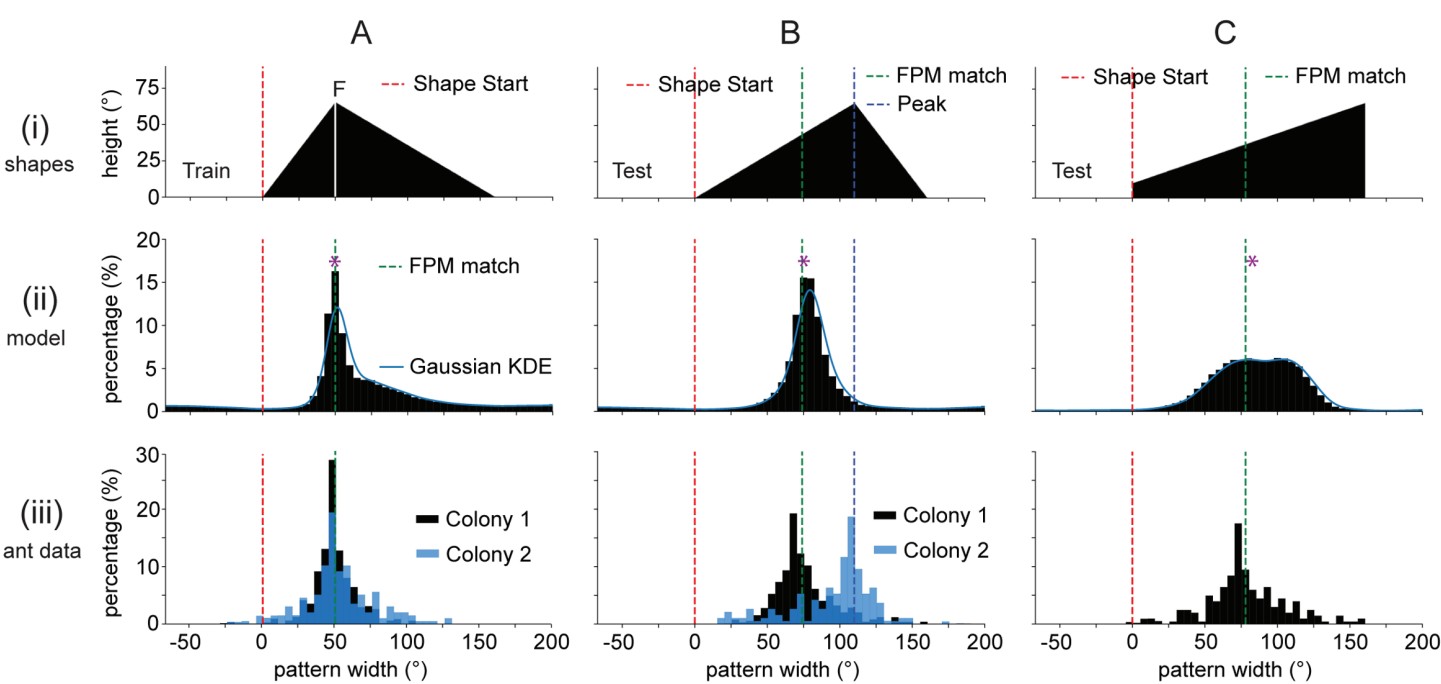

**Fig 4. Results after training with a triangular shape.** Column (A). The train shape for the second set of experiments consists of a scalene triangle with width 160° and height 65°, and summit 50° to the right of the shape starting point. Column (B). The first test image is the train image reflected horizontally. The blue dotted line represents the location of the peak. Column (C). The second test image is a trapezoid of width 160°, with heights of 10° and 65° on the left and right sides respectively. Row (i). Schematic depiction of the train and test images. Row (ii). Simulated saccade-end-point distributions. The purple asterisk denotes the minimum of the Image Difference Function (IDF). Row (iii). Experimental data inferred from [22]. For the train shape and first test shape, the data for two different colonies is displayed (Colony 1 and Colony 2) with colours black and blue respectively. Other figure conventions as in Fig 3.

2) is centred below the peak of the triangle instead of the FPM-match direction. Whilst the results clash with the outputs of our computational model displayed in Fig 4B(ii), these experimental results admit a natural interpretation within this modelling framework. If we imagine train and test shapes aligned on their peaks, then the two views are horizontal mirror reflections of each other around the centre line. As a consequence of this, the comparison of the train and test shapes in the left and right hemispheres result in equal matches, regardless of which method one chooses to compare the shapes. So, if the ants only memorise a single view from the centre of the arena, the peak would define the angle where we expect left and right novelties to balance.

Indeed, when we test the model after training solely on the first image in the sequence (as experienced from the centre of the arena), the novelty difference curve displays a zero-crossing at the peak-defined direction for a variety of parameter settings (S4A Fig row (iv)). As we train the model on increasingly long portions of the path (S4B and S4C Fig) the location of the zero-crossings shifts leftwards towards the FPM-defined direction. On the other hand, the novelty sum does not show a qualitative change in its outputs as we change the length of the training path (S4 Fig row (iii)), indicating how it is the novelty difference that can explain the behavioural differences. The model-derived distributions in the different training regimes show an increasing certainty in the FPM-defined direction as the length of the training path increases (S4 Fig row (iv)). Overall, the outputs of the model corroborate a speculative explanation for the different colony behaviours based on the length of the path that is memorised.

## Results for composite shapes, a tentative explanation of segmentation

Further experiments in [22] consisted of training with a composite shape, made up of two abutting triangles (Fig 5). Across three sets of experiments (I, II and III in Fig 5) the same training shape is used with different feeder locations. Specifically, the feeder location shifts rightward from I to II to III. The experimental results for this set of shapes can be interpreted as indicative of a visual shape segmentation mechanism (another higher-order processing of the visual scene) when the feeder direction is close to the centre of a component of the composite shape, because the ants aim for the appropriate FPM of a test shape as if they had only learned the FPM of a single component of the training shape.

Indeed, when ants are trained with a shape composed of two parts (e.g., two triangles), there are two possible ways that an FPM value can be computed. The first is that the FPM is computed over the whole visual scene composed of two triangles, which we refer to as the whole FPM (wFPM). The second is that the FPM is computed over a segmented part of the scene (one triangle), which we refer to as the segmented FPM (sFPM). As observed in column (B) of Fig 5, when ants are trained to a feeder closer to the left of the composite shape, and tested with a wide rectangle, the goal direction distributions are centred around the sFPM (Fig 5IB and 5IIB). Assuming ants are guided by an FPM value, this is indicative that the ants learned about heading directions relative to the leftmost triangle only, therefore suggesting a shape segmentation mechanism (or attention-like process, see Discussion). On the other hand, if ants are trained to a feeder close to the centre of the composite shape, the goal direction distribution with the test shapes are centred at the wFPM (Fig 5IIIB), which in this case suggests no segmentation took place.

The outputs of the computational model for this set of shapes show indicators that assuming low novelty and lateral balance may be sufficient in explaining an apparent segmentation of the scene. As can be seen in (Fig 5IB and 5IIB), the simulated modes shift away from the wFPM for this set of shapes in the direction of the sFPM. On the other hand, in (Fig 5IIIB),

 

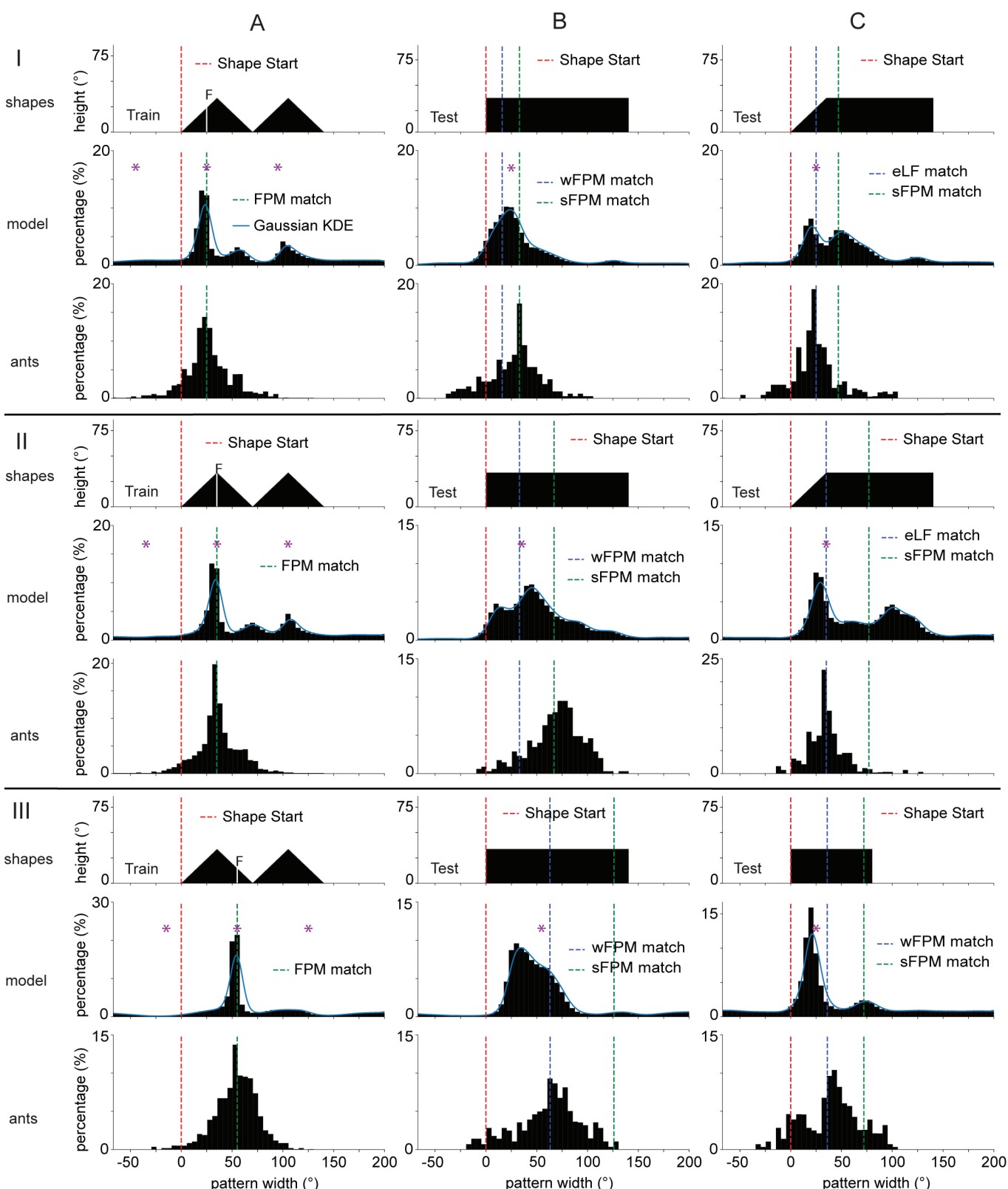

**Fig 5. Results after training with composite shapes.** This figure displays the results for three separate sets of experiments (I, II and III) where the training location is at different places within the same composite shape. Set I. The training shapes are a pair of abutting triangles, each of which is isosceles with base 70° and height 35°. The feeder location is 25° rightwards of the shape start location. Set II. Same training shape as in previous set but the feeder is located at the

 

centre of the first triangle. Set III. The feeder is located 55° rightwards of the shape start location. Column (A). The results for the train shapes tested against themselves. Column (B). The test shape for all three sets is a rectangle 140° wide and 35° high. The blue dotted line represents the wFPM-match direction (FPM-match direction if we assume the FPM is computed across both triangles in training). The green dotted line represents the sFPM-match direction (FPM-match direction if we assume the FPM is computed across a segmented part of the image). Column (C). For sets I and II, the test shape is a trapezoid with width 140°, height 35°, and the top left vertex 35° rightwards of the start of shape. For these sets, we display a vertical dotted line at the eLF-match direction (direction where left slanted edge matches its retinal position between train and test runs). We also show a green dotted line at the sFPM-match direction, as described for previous set. For set III, the test shape is a 80° wide and 35° high rectangle. Across all examples, the purple asterisk(s) highlight the local minima of the rotational IDF. Other figure conventions as in Fig 3.

the mode is closer to the wFPM than the sFPM, as expected from the experimental results. Given that these are "raw" results obtained with no parameter tuning, the assumptions used in this modelling work have potential in capturing certain aspects of the experimental outcomes. Still, given the inexact match between model and experiments, we do not exclude that further computational steps are needed to fully capture the results (see Discussion).

Interestingly, the model predicts a mode at the correct location in (Fig 5IC and 5IIC), without having to resort to edge (or feature) matching, which was the explanation given in the original paper [22]. Over all sets of examples, the pooled distributions obtain an average distance between modes of 11.6° ± 11.5° (mean ± std). When studying the performances over all shapes and parameters separately we obtain a distance between modes of 10.1° ± 9.3° (mean ± std). The average performance against parameter values are displayed in (S2C Fig), showing that over a broad range of parameter values, the outputs of the computational model are not inconsistent with the experimental findings.

## Discussion

Visual memory based navigation in insects has been extensively studied at a behavioural level [2–5,40], in models [9,12,14,15], and in terms of its underlying neural substrates [16–18,41,42]. Whilst it is clear that ants collect and use visual memories to navigate their environment, it is still largely unknown what aspects of a visual scene are memorised and how the memories are used to affect steering decisions [20,21,33,43]. Experiments have shown that the wood ant *Formica rufa* can learn a heading direction using the proportion of a large shape that lies to the left of its visual centre in training [22], a proportion referred to as fractional position of mass (or FPM for short). This suggests that ants can memorise higher-order features extracted over the whole extent of a visual scene. In this work, we used a simple model leveraging known neuroanatomy and connectivity in ants [35–37] to explore whether navigation to the FPM could be an emergent property of the bilateral structure of the visual and memory circuits. In so doing, we assumed a retinotopic image representation, and showed how the FPM of a visual scene can be implicitly encoded in the memory dynamics of the bilateral network. This work therefore demonstrates that some higher order visual processing and visual cognition could be emergent properties of the overall neural architecture rather than outputs of discrete visual processing modules.

### What visual processing is underpinning visual navigation in ants?

It is still unknown what visual information is used by ants in visual memory based navigation. At a neural level, visual information reaches the MBs via the anterior superior optical tract (ASOT) which relays information from each visual field to both MBs. Neuroscientific studies [35–37] reveal extensive projections from the Medulla (ME) of the Optic Lobes (OL) to the MBs, and a smaller amount of projections from the Lobula (LO). The ME and LO are known to specialise respectively in small and wide field information and can convey chromatic,

temporal, and motion features in bees [44]. A range of experimental studies [22,24–27,45] highlight that insects appear to use higher-order features extracted over the whole visual scene, such as the centre of mass (CoM) or fractional position of mass (FPM) of large shapes. However, many of these results can be explained by the present study, or by other models showing higher-order parameters emerging from a population of simple visual filters [46].

The success of this modelling approach relies upon the assumption that visual processing preserves sufficient retinotopy so that ratios are roughly conserved from retinal to vPN layers (see theoretical section of results). To clarify, we do not need the vPNs to have an exact topographic representation of the visual scene, but to preserve a statistical representations of the retinal information. In this sense, this approach may be stable against processing by different kinds of filters which may occur within the ME and LO of the Optic Lobes, which we can assume to preserve a kind of semi-retinotopic representation. On the other extreme of the representation spectrum are approaches that are composed solely of features in the global spatial frequency domain, such as rotationally invariant features based upon Zernike moments [28,29]. We do not expect our modelling approach to work if we apply such processing because this type of representation would not preserve the statistics of the retinotopic representation.

In modelling, we further assumed that the visual Projection Neurons (vPNs) respond to the black parts of the visual scene which defines the shape. This is consistent with experimental and modelling papers that suggest ant photoreceptor tuning might pre-process visual scenes into objects that contrast against sky [47]. The fundamental importance of the panoramic skyline (objects appearing as dark regions in a scene contrasted against the bright sky), is shown experimentally in [3]. Similarly robot studies have shown the importance of sky segmentation (based on UV light) for robust navigation across different lighting conditions [48,49]. Together, these studies suggest that the early stages of the ant's visual system may be optimised for separating the sky from ground, leading to a more reliable image representation (e.g., invariant to weather patterns).

### Are the results with composite shapes indicative of a selective attention mechanism?

The modelling assumptions of the present study partially helped in explaining the experimentally obtained heading directions for composite shapes, which hinted at a visual segmentation mechanism. However, it is possible that further dynamics are needed to fully explain these results. One such candidate is that ant vision is endowed with an attention-like process [50]. Whilst attention is a concept more commonly associated with primates [51,52], attention-like processes have also been demonstrated in several insect species [50,53]. Visual selective attention plays a fundamental role in tracking, whereby a salient target can suppress the responses of other targets, as shown in e.g., praying mantises [54], fruit flies [55] and dragonflies [56]. However, it is not known whether such attentional processes would also be used in visual navigation.

If the ant visual system selectively favours information coming from a structurally cohesive and central part of the visual scene, then the visual memories could be limited to one of two triangles explaining an effective segmentation of the visual scene. As mentioned above, this upward weighting of one triangle over the other could be the result of a selective attention mechanism. However, we can also imagine obtaining similar results from an architectural mechanism, for instance by having higher density of projections attributed to the central parts of the visual fields. Another part of MB dynamics that was not considered in our modelling work is the existence of recurrent connections at the Kenyon Cell layer [57]. These

connections have the potential to complexify the memory formation process, for instance they have been shown to allow sequence learning [58] and point attractor dynamics [59]. It is possible that visual information undergoes a transformation due to the KC to KC recurrent connections which dampens peripheral information, providing another speculative explanation for the effective segmentation of the visual scene.

### How do the MBONs drive downstream steering behaviours?

In this modelling work, we explored whether a signal for the FPM-match direction exists within the dynamics of bilateral memories. In so doing we did not make assumptions as to how that signal may be used downstream to drive behaviour, and therefore the model could be used in a range of visuo-motor control strategies. However, considering the specific behaviour that was used in the original paper to measure goal directions can lead to interesting speculations into its neural mechanism. The goal directions in [22] are measured using the end-points of saccade-like-body turns (SLTs), which are periods of high rotational velocity shown in [39] to be visually guided turns correcting the heading direction towards a goal. The fact that the angular speed of saccades is proportional to their angular error at the beginning of the turn [39] is a strong indicator that ants have an internal estimate of turn size prior to initiating saccades. This finding shows striking similarities with the recently discovered function of PFL2 cells in *Drosophila melanogaster*, which have been demonstrated to adaptively tune the speed of turns based on the angular difference between a goal direction and the current heading direction maintained within the Central Complex [60]. The integration of contextual (body-centric) information with a world-centric reference frame (and vice versa) has been shown to rely upon the Fanshaped Body (FB) of the Central Complex (CX) in numerous studies [60–64]. In light of these facts, we postulate that in ants, Mushroom Body outputs (MBONs) are used to set a goal direction within the FB of the CX. Because of the conclusions of our present study, we speculate that goal directions are set when the novelty is (comparatively) low and also balanced across left and right hemispheres. Saccades would then constitute turns that correct the current heading direction towards this set goal direction.

The fact that SLTs occur at a specific phase [65] within the intrinsic oscillatory cycle of the ants [66], further suggests a recurrent signal from the Lateral Accessory Lobes (LALs) is implicated in initiating the turns. This fits well with the dense connectivity patterns between the FB and LALs forming functional loops [67,68]. We further note that since the turns occur at the outer edge of the oscillations [65], the difference between the current heading and the goal direction constitutes a prediction error (this difference should be zero if the ant is on track). Therefore, it seems that SLTs constitute a correction mechanism built on top of other navigational strategies.

### Conclusion

The major modelling insight and prediction of this study, is that a bilateral brain architecture allows ants to balance the quality of their memories in left and right hemispheres. As shown in this work, the balancing of memory valences across hemispheres could allow the retrieval of higher-order properties of a shape in simple visual settings (e.g., FPM). In particular, the FPM has a "scale-invariant" quality (remaining the same at different distances to an object) which might confer benefits when navigating to a landmark. It is also possible that balancing two separate assessments of a panoramic view contributes robustness to the resultant navigational behaviours [32]. Still, we do not know the full extent of how the bilateral brain architecture contributes to navigational functions, which defines a promising avenue for future research.

## Methods

### Image geometry

The experimental setting of [22] consisted of an arena of radius $R$ = 1.5m within which ants were trained to head to a feeder $r$ = 0.6m away from the centre (Fig 1A). The stimulus angles of the shapes used (as measured from the centre of the arena) are given in the original paper, and inferred where exact values are not given. Images are exported with vertical and horizontal axes corresponding respectively to polar and azimuthal angles. We represent the black colour with pixel value 1 and white with pixel value 0, allowing for greyscale values in between.

The equations for how the experienced shape transforms as a hypothetical ant moves within a circular arena were determined using trigonometry. Let $Z$ be the point at the base of the arena wall representing azimuthal angle 0°. Let $\theta$ and $\phi$ be the azimuthal and polar stimulus angles of an arbitrary point $P$ on the arena wall, as measured from the centre $O$ of the arena (Fig 1A). If $L$ is a new location within the arena and $P_{xy} = (R\cos(\theta), R\sin(\theta))$ is the location of point $P$ projected onto the base of the arena, then the angles transform as follows:

$$\theta' = \cos^{-1}\left( \frac{\overrightarrow{LZ} \cdot \overrightarrow{LP}}{|LZ||LP|} \right)$$

$$\phi' = \tan^{-1}\left( \frac{R\tan(\phi)}{|LP|} \right)$$

By transforming all points constituting the outline of a shape, we export (90 by 360) greyscale images representing the visual pattern as experienced at required locations in the arena.

### Image processing

From an exported image (panoramic), we extract two images corresponding to the left and right visual fields, according to an overlap parameter (Fig 1B). We do so by cropping the azimuthal angles to the range $\left[ \frac{overlap}{2}, 180 + \frac{overlap}{2} \right]$ for the left visual field, and range $\left[ 180 - \frac{overlap}{2}, 360 - \frac{overlap}{2} \right]$ for the right visual field. The two images are then stacked horizontally creating one feature-image with two implicit parts. To better capture the resolution of ant vision, we reduce the image resolution by cropping the top 10 layers of pixels, and downsampling the resulting image by a factor of 4. The downsampling is done by taking the mean of each 4 by 4 block in the (80 by 360) cropped image, resulting in a (20 by 90) image with 4° per pixel resolution.

The training paths are created assuming an ant moves directly from the centre of the arena towards the feeder location, facing straight ahead. We export $N_{train}$ = 30 images for each training pattern taken in equal steps from distance $r$ = 0m to $r$ = 0.6m. Given that we cannot assume in advance a movement direction for simulated trajectories, we only use $N_{test}$ = 1 image taken from the centre of the arena for the test shapes.

### Network architecture

We model the Mushroom Bodies with an artificial neural network (ANN), represented pictorially in (Fig 1B). The model's implementation details are most similar to those used in [38]. The network is composed of three layers with four types of neurons:

- The first layer is the input image corresponding to the left and right visual fields stacked horizontally, which is a matrix of shape (20, 90). The Visual Projection neurons (vPNs) fire with strength given by the greyscale value of the corresponding pixel ranging between 0 (white) and 1 (black).
- The second layer is composed of two Mushroom Body (MBs) compartments, each having $N_{KC} = 25,000$ Kenyon Cell (KC) neurons, which is representative of numbers found in some insects (see e.g., [36]). The input strength of a KC is computed as a weighted sum over connected pixel values $\text{input}(KC) = \sum_x w_{xKC} \times x$. Where $w_{xKC}$ denotes the weight of the connection from pixel $x$ to the given KC (as described in next section).
- Each MB compartment is also fully connected to an Anterior Paired Lateral neuron (APL), which induces a normalisation on the overall activity of the MB [69]. For a given pattern of firing Kenyon Cells, the APL will select the proportion $p$ of cells with the strongest activity and allow them to fire (output is set to 1), all other cells do not fire (output is set to 0). In essence, this binarises KC activity and ensures a sparse representation.
- The third layer is composed of two Mushroom Body Output Neurons (MBONs), which fire as a weighted sum of KC outputs: $\text{output}(MBON) = \sum_{KC} w_{KC} \times \text{output}(KC)$. The connections and weights between KCs and MBONs are described in the next section.

## Network connections

The layers are connected and weights are set as follows:

- A sparse pattern of connections is created between vPNs and KCs so that KCs have on average $K$ connections with vPNs. The connections are generated by picking $\frac{2KN_{KC}}{N_{vPN}}$ Kenyon Cells for each vPN (separately for left and right visual fields) according to the following bias: probability $1-crossover$ of connecting to the ipsilateral MB, and probability $crossover$ of connecting with the contralateral MB.
- We consider two modes of weight initialisation. In the first case (constant weights), all vPN to KC connection weights are initialised to $w_{xKC} = 1/K$. In the second case (random weights), each weight is assigned a random and uniform value in the range $[0, \frac{2}{K}]$. The range is chosen so that weights have a mean value of $1/K$ in both cases.
- Each MB is fully connected to the MBON ipsilateral to it. Connection weights are initialised to $w_{KC}^{\text{initial}} = \frac{1}{pN_{KC}}$, chosen to have a maximal expected novelty value of 1. As with the previous case, we have a random mode of weight initialisation where each output weight is assigned a random and uniform value in the range $[0, \frac{2}{pN_{KC}}]$. These weights are subsequently updated during training, as explained in the next section.

## Network training

The training acts solely on the connections from Kenyon Cells to MBONs, and works by depreciating the weights from KCs that fire for the train image(s). This type of update is often referred to as "Anti-Hebbian" despite no explicit referral to the postsynaptic neuron's firing. In Hebbian learning, neurons that fire in close temporal proximity would have their synaptic weights strengthened (in Anti-Hebbian, synaptic weights are weakened). In our model, the action of Dopaminergic neurons (DANs) and the post synaptic neuron's firing (MBON) at train time are not explicitly modelled, making for a simpler learning rule which considers only Kenyon Cells. Each time a Kenyon Cell fires in the training phase, its initial output weight $w_{KC}^{\text{initial}}$ is multiplied by a factor of $\alpha < 1$.

The weight update for the whole training can be condensed into a single formula. In effect, each weight is multiplied by a factor of $\alpha^{n_{tot}}$ where $n_{tot}$ is the total number of times the KC fired in training. In equations:

$$w_{KC}^{\text{trained}} = w_{KC}^{\text{initial}} \times \alpha^{n_{tot}}$$

We use $\alpha = 0.95$ as the default value and study the effect of this parameter in (S3 Fig).

## Network testing

To test a network with a single image, we forward-pass the image through the trained network and take the output of the MBONs (Fig 1B) as measures of (left and right) novelties. To produce rotational signals, we forward-pass the image associated with each of the 360 possible facing directions. With two MB outputs, there are four signals that we consider: *left novelty*, *right novelty*, *novelty sum*, *novelty difference* (S1 Fig). The novelty sum and difference are used to produce distributions of goal directions, as described in the next section. Rotational signals are displayed as the mean of $n_{models} = 50$ model initializations, with a shaded area spanning one standard deviation either side of the mean (S1 and S3 Figs).

## Generating heading distributions

From the rotational novelty difference and rotational novelty sum signals (S1F and S1D Fig), we extract distributions that model the distribution of ant saccade-like-turn (SLTs) end points from [22]. To produce a distribution from the outputs of our computational model, we take a Monte Carlo approach. The method assumes that both proximity to the minimum of the novelty sum and proximity to a zero-crossing of the novelty difference are important for determining goal directions. We generate tentative directions uniformly at random in range $[-180°, 180°]$ and accept with probability:

$$P(S, \Delta) = e^{-\beta \frac{|S - S_{min}|}{|S_{max} - S_{min}|}} \times e^{-\beta \frac{|\Delta|}{|\Delta|_{max}}}$$

Where $S$ indicates the novelty sum for that given direction, and $\Delta$ is the novelty difference. $S_{max}$, $S_{min}$ are the maximum and minimum values of the novelty sum, $|\Delta|_{max}$ is the maximum absolute novelty difference (see S1D and S1F Fig). The value of the exponential decay $\beta$ tunes the variance of the simulated distributions, and we use $\beta = 4$ to qualitatively match the variance of the distributions in [22]. Histograms are produced by binning the data in range $[-70, 200]$ with a bin width of $5°$, to match the presentation of results in [22].

## Pooling heading distributions

Given a desire to be sure that our results are derived mainly from the architecture of the model, rather than esoteric parameter choices, we generate the aggregate behaviour of the model by pooling results over a range of parameter settings. Parameters $\alpha = 0.95$, $K = 8$, $p = 0.05$ are kept fixed at their default values because they have little effect on model outputs as far as zero-crossings are concerned (S3 Fig), and we systematically study all 30 pairs of *crossover* and *overlap* in the ranges $[0, 0.1, 0.2, 0.3, 0.4]$ and $[0°, 8°, 16°, 24°, 32°, 40°]$ respectively. The range of overlap values was chosen based on the experimentally observed anatomy of *Cataglyphis sp.* in [70].

For each of the 5 *crossover* parameter values, 50 models are initialised. Each of the 50 models is trained in turn with each of the 6 visual *overlap* parameter values. The resulting

$50 \times 30$ model and training combinations are independently tested to produce rotational novelty difference and rotational novelty sum signals, which are used to simulate $n_{saccade} = 100$ SLT end points each (as described in previous section). We therefore pool together $n_{saccade} = 100$ saccade end points for $n_{repeat} = 50$ model initialisations and $n_{params} = 30$ possible pairs of parameter settings, resulting in a total of $n_{montecarlo} = 150,000$ simulated saccades which we use to infer a distribution.

## Distribution modes

To infer mode locations for the simulated distributions, we use a continuous estimate of our predicted distributions. This step ensures that modes extracted from the predicted distributions (which can be bimodal) convey broad trends rather than small sample artefacts. We produce such a smoothed approximation using Kernel Density Estimation (KDE) with a gaussian kernel, and a bandwidth of $h = 5°$ to match the angular acuity of our data. We extract the KDE estimate $\hat{f}(y)$ in steps of $1°$ and define a point $y$ to be a mode if it is a local optimum of the KDE function. To avoid duplicate optimums, we ensure a local optimum is preceded by an increase for 5 degrees and succeeded by a decrease for 5 degrees.

## Distribution performance

For a simulated distribution, we extract a measure of performance (which we refer to as the "distance between modes" DBM) using the minimum distance between its modes and the experimentally observed mode, inferred from [22]. In equations

$$DBM = \min_i \left| m_i - m^{\text{true}} \right|,$$

where $m_i$ are the modes extracted from the simulated distribution and $m^{true}$ is the true mode from the original experiments. We always repeat performance estimations $n_{iter} = 10$ times and average results to mitigate for variance. When we compute a performance over $N$ experiments, we use the Mean Absolute Error (MAE) of the predictions for the individual experiments:

$$MAE = \frac{1}{N} \sum_{j=1}^{N} \min_i \left| m_{ji} - m_j^{\text{true}} \right|$$

The performances are computed independently for the pooled distributions and for all combinations of *crossover* and *overlap* parameter values (S2 Fig).

## Model parameters

The model parameters are listed below along with their function:

- $\alpha$: the learning rate of the network. Controls how fast weights decrease in training.
- $K$: expected number of connections of Kenyon Cells. Controls the sparsity of the network.
- $p$: the proportion of KCs that are allowed to fire. Controls the normalisation of MB activity.
- *crossover*: the probability that a connection crosses over from one visual field to the contralateral MB. Controls the ipsilateral bias of the network initialisation.

**Table 1. Default parameter values and ranges studied.**

| Parameter | Default Value | Range Studied |
|---|---|---|
| $\alpha$ | 0.95 | 0.9, 0.95, 0.99 |
| $K$ | 8 | 4, 8, 16, 32 |
| $p$ | 0.05 | 0.025, 0.05, 0.1, 0.2 |
| crossover | 0.2 | 0, 0.1, 0.2, 0.3, 0.4, 0.5 |
| overlap | 0° | 0°, 8°, 16°, 24°, 32°, 40° |

- *overlap*: the amount of angular overlap of left and right visual fields. Changes the azimuthal cutoffs of left and right fields of view.

When a parameter is not explicitly studied, its value reverts to the default given in Table 1. The effect of parameters on model outputs is explored in (S3 Fig).

## Supporting information

**S1 Fig. Signals from the bilateral memory network.** (A) The train image used for this example is a rectangle of width 160° and height 38° with feeder inset 30° from the left side of shape (Fig 2A). (B) The test shape is a rectangle of width 80° and height 38° (Fig 2B). (C) The left novelty is plotted as a function of angular orientation, with 0° being the direction to the left edge of the train and test shapes. (D) The sum of left and right novelties is plotted. $S_{min}$ denotes the minimum value of the signal. $S_{max}$ denotes the maximum value of the signal. $S$ denotes the value of the signal at an angular orientation of 75°. (E) The right novelty is plotted as a function of angular orientation. (F) The difference of left and right novelties is plotted against angular orientation. $\Delta_{max}$ denotes the maximum novelty difference in absolute value. $\Delta$ denotes the novelty difference at an angular orientation of 75°.
(TIF)

**S2 Fig. Model performance heatmaps.** Heatmaps of average performance of the model, denoted as DBM ("distance between modes") for all combinations of crossover [0, 0.1, 0.2, 0.3, 0.4] and overlap [0°, 8°, 16°, 24°, 32°, 40°] parameter values shown for (A) First set of experiments. (B) Second set of experiments. (C) Third set of experiments. (D) Overall.
(TIF)

**S3 Fig. Parameter scans for simple train-to-test comparisons.** The train image used for all results in this figure is a rectangle of width 160° and height 38°. Column (A) consists of testing on a 120° by 38° rectangle. Column (B) consists of testing on a 80° by 38° rectangle. Column (C) consists of testing on a 40° by 38° rectangle. Row (i) consists of rotational novelty difference plots for different values of parameter $\alpha$ in the range [0.9, 0.95, 0.99]. Row (ii) studies parameter $K$ in the range [4, 8, 16, 32]. Row (iii) studies parameter $p$ in the range [0.025, 0.05, 0.1, 0.2]. Row (iv) studies parameter *crossover* in the range [0, 0.1, 0.2, 0.3, 0.4, 0.5]. Row (v) studies parameter *overlap* in the range [0°, 8°, 16°, 24°, 32°, 40°]. Row (vi) studies the model with randomly initialised connection weights. In the first case we initialise vPN to KC weights uniformly in $[0, 2/K]$. In the second case KC to MBON weights are uniformly distributed in $[0, 2/pN_{KC}]$. In the third case, both are initialised randomly as described above.
(TIF)

**S4 Fig. Length of training path affects results for reflected triangle.** (A) Training only on first view (as experienced from centre of the arena). (B) Training up until halfway to the feeder location ($N_{train}$ = 15). (C) Training on the whole path ($N_{train}$ = 30). (i) The training

shape as it is experienced from the last training location. (ii) The test shape. (iii) The novelty sum output of the trained model after testing. We display the curves for 5 different parameter settings (crossover, overlap = [0,0°], [0.1, 8°], [0.2, 16°], [0.3, 24°], [0.4, 32°]) (iv) The novelty difference output of the model. (v) The estimated distributions (pooled) for all three modes of training.
(TIF)

**S1 Appendix. Additional derivations.** We include additional mathematical derivations supporting the statements made in the second section of the results.
(PDF)

## Author contributions

**Conceptualization:** Giulio Filippi, Paul Graham.

**Formal analysis:** Giulio Filippi.

**Supervision:** James Knight, Andrew Philippides, Paul Graham.

**Writing – original draft:** Giulio Filippi.

**Writing – review & editing:** Giulio Filippi, James Knight, Andrew Philippides, Paul Graham.

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
