## [Decision Letter · Decision Letter 0]

10 Mar 2025

PCOMPBIOL-D-24-02038

Lateralised memory networks explain the use of higher-order visual features in navigating insects

PLOS Computational Biology

Dear Dr. Filippi,

Thank you for submitting your manuscript to PLOS Computational Biology. After careful consideration, we feel that it has merit but does not fully meet PLOS Computational Biology's publication criteria as it currently stands. Therefore, we invite you to submit a revised version of the manuscript that addresses the points raised during the review process.

Please submit your revised manuscript within 30 days May 10 2025 11:59PM. If you will need more time than this to complete your revisions, please reply to this message or contact the journal office at ploscompbiol@plos.org. Please include the following items when submitting your revised manuscript:

We look forward to receiving your revised manuscript.

Kind regards,

Matthieu Louis

Academic Editor

PLOS Computational Biology

Lyle Graham

Section Editor

PLOS Computational Biology

**Journal Requirements:**

2) Some material included in your submission may be copyrighted. According to PLOSu2019s copyright policy, authors who use figures or other material (e.g., graphics, clipart, maps) from another author or copyright holder must demonstrate or obtain permission to publish this material under the Creative Commons Attribution 4.0 International (CC BY 4.0) License used by PLOS journals. Please closely review the details of PLOSu2019s copyright requirements here: PLOS Licenses and Copyright. If you need to request permissions from a copyright holder, you may use PLOS's Copyright Content Permission form.

Potential Copyright Issues:

i) Figure 1A. Please confirm whether you drew the images / clip-art within the figure panels by hand. If you did not draw the images, please provide (a) a link to the source of the images or icons and their license / terms of use; or (b) written permission from the copyright holder to publish the images or icons under our CC BY 4.0 license. Alternatively, you may replace the images with open source alternatives. See these open source resources you may use to replace images / clip-art:

**Reviewers' comments:**

Reviewer's Responses to Questions

**Comments to the Authors:**

**Please note that one of the reviews is uploaded as an attachment.**

Reviewer #1: This study showcases an elegant algorithm that lays out, how a processing of higher visual features may emerge by considering the hemispherical structure of the insect brain/mushroom bodies. Within the context of visual navigation in ants, this lays an important step considering visual features other than the holistic usage of the entire panorama.

Here are my comments:

Please add a more detailed description of your code to the readme.

LL. 20-25: Just for the context within the range of naturally occuring behaviours in ants, where does the Lent study fit into the foraging process? Would this be mostly relevant for a "final approach", where the ant has almost reached the feeder and wants to find the very specific location? Or would this FPM also reflect a mechanism which could be applied at any section during route following, from the nest to the food location?

LL. 191-198: The test shapes cannot be contained in the train shapes, because they do not have the same relative shape? For example, the Scalene shape triangle in 4B would only be contained in the train shape if it were, for example, scaled down by 20% in the x and y?

LL. 327-336: Attention in Human EEG studies show increased activity in the hemisphere associated with the processing of the task relevant stimulus (Antonov et al 2020, "Too little, too late, and in the wrong place..."). In the context of this model, can you imagine a mechanism which up-/downregulates the processing of the task relevant segments?

Furthermore, in visually complex scenes there will be many more shapes in the visual fields. Would this mechanism work with learnt shapes covering the entire panorama, or would this need some selective attention mechanism which depresses the computation of non-relevant shapes?

L. 388: Maybe change "view" to "panoramic view"? So that the distinction between monocular and binocular is stronger.

L. 436: Why Nkc = 25,000? In total 50,000. Other studies use less KCs for representing the whole MB. If you used similarly lower numbers, would that negatively effect the results of your model, and if so, how?

L. 455: The crossover probability covers the entire contralateral vPN space, or only the space minus overlap?

L. 514: The crossover is from the contralateral "eye"? In other words, 0-50% random pixels from the contralateral retina?

S1 Fig: Above the plots, maybe show the training image like in S3.

And lastly, i want to recapitulate the processing steps as i understood them; due to the manuscript-structure, the individual, but consecutive steps were a little too distributed for me. If there is anything missing/wrong in my recapitulation, maybe point out these steps more in detail.

1. Generation of images from the centre of the scene moving straight towards the feeder, so that the feeder would always be in the centre/origin of the image. These images represent the route memories.

2. Training the MB-Model using the route memories.

3. Presenting test-images to the model with different rotations, so that a novelty function for (most of) the panorama can be generated. For each shape, were these also presented from different places on the route, just like route memories?

4. Applying the Monte Carlo method to generate the most likely heading directions, dependent on the novelty sum and novelty difference.

5. Comparison with the behavioural data.

Reviewer #2: see attachment.

Reviewer #3: The authors revisit here the question of whether ants are able to use one specific higher-order visual feature, the "fractional position of mass" (FPM), to navigate. While this may seem like a rather niche or narrow question, the methodology shown in this work and its general message are very significant to the field: By applying a model constrained by the known anatomy and visual information processing in the ant brain, the authors demonstrate that encoding and recalling of an otherwise thought "higher-order" kind of information can actually be a purely emergent property of the brain's bilateral organization, rather than a specifically tuned, dedicated module. Their model reproduces the experimental findings from the Lent et al 2013 paper quite remarkably. I also appreciate that the authors contextualize their model of the Mushroom Bodies within the broader context of areas involved in navigation, the Central Complex and Lateral Accessory Lobes (paralleling the recently uncovered role of PFL2 cells in Drosophila).

The approach shown in this paper and its main conclusion (that "a bilateral brain architecture allows ants to balance the quality of their memories in left and right hemispheres when navigating") clearly fit whithin the current state of research in the neurobiology of insect navigation, and represent in my opinion an overall coherent and convincing example of how modelling allows us to draw the link between complex behaviour and brain connectivity, making this work a strong contribution to the field.

Some comments for the authors:

"However, if the memory of the path were biased towards the first few views (due to some unknown mechanism), then we would expect that left and right novelties equate at the peak-defined direction, providing a tentative explanation for the bifurcation in colony behaviors." (l. 226 - 229)

It would maybe be beneficial to quantify the change in information received by the ommaditia between two steps in the simulation, as one would expect a stronger change in the signal between two steps when the angular size of the object is greater (i.e. a smaller change in the familiarity between two steps at the start of the route compared to the end).

Also, including some model results using more complex visual scenes and/or a more complex insect eye model might generate some very interesting predictions to be tested experimentally.

**Have the authors made all data and (if applicable) computational code underlying the findings in their manuscript fully available?**

Reviewer #1: Yes

Reviewer #2: None

Reviewer #3: Yes

PLOS authors have the option to publish the peer review history of their article (what does this mean?). If published, this will include your full peer review and any attached files.

Reviewer #1: **Yes: **Fabian Steinbeck

Reviewer #2: No

Reviewer #3: No

**Figure resubmission:**
---

## [Decision Letter · Decision Letter 1]

15 May 2025

PCOMPBIOL-D-24-02038R1

Lateralised memory networks may explain the use of higher-order visual features in navigating insects

PLOS Computational Biology

Dear Dr. Filippi,

Thank you for submitting your manuscript to PLOS Computational Biology. While we feel that it has strong merit, it does not fully meet PLOS Computational Biology's publication criteria as it currently stands. Therefore, we invite you to submit a revised version of the manuscript that addresses the final set of points raised during the review process.

Please submit your revised manuscript within 30 days Jul 15 2025 11:59PM. If you will need more time than this to complete your revisions, please reply to this message or contact the journal office at ploscompbiol@plos.org. Please include the following items when submitting your revised manuscript:

We look forward to receiving your revised manuscript.

Kind regards,

Matthieu Louis

Academic Editor

PLOS Computational Biology

Lyle Graham

Section Editor

PLOS Computational Biology

**Reviewers' comments:**

Reviewer's Responses to Questions

Reviewer #1: All my comments have been adressed appropriately and improved my and any reader's understanding.

Reviewer #2: The authors have significantly revised the paper and almost all of my previous comments have been addressed, except for two minor issues:

1. Regarding the clarification on tropotaxis: frankly I'm more confused as the response seems to distinguish 'using difference in bilateral signals' (tropotaxis as understood by the authors) vs 'balancing bilateral signals' (the approach of this paper). Technically/Mathematically, I don't see the difference between these two descriptions, and the paper talks about left-right difference being zero, seemingly interchangeably with left-right being balanced.

2. In the new supmaths5.pdf:

(2a) Right before Equation (S5), please define \sigma clearly. I guess whoever reads this part will probabily guess it to be standard deviation, but of which distribution? Or, is it just a generic notation as \mathbb{E}?

(2b) Right before Equation (S13), the assumption of the base rates X/T and Y/T being small is made. I'd appreciate it if the authors can elaborate more this assumption.

There are also some other very minor issues such as spelling consistency issue (e.g., 'color' instead of 'colour' as the authors stated they want British spelling), which I don't think hinder the paper's quality at all. Similarly, I think the revised paper fulfills the criteria for publication, even not addressing the minor points I raised above, except for (2b), because PNs are not well known for sparse activation (as KCs). I suggest the authors making it clearer if this particular assumption is made base on biologically evidence or modelling assumption, or just for mathematical convenience -- All these options are acceptable in my opinion.

**Have the authors made all data and (if applicable) computational code underlying the findings in their manuscript fully available?**

Reviewer #1: Yes

Reviewer #2: Yes

PLOS authors have the option to publish the peer review history of their article (what does this mean?). If published, this will include your full peer review and any attached files.

Reviewer #1: **Yes: **Fabian Steinbeck

Reviewer #2: No

**Figure resubmission:**
---

## [Editor Report · Decision Letter 2]

24 May 2025

Dear Mr Filippi,

We are pleased to inform you that your manuscript 'Lateralised memory networks may explain the use of higher-order visual features in navigating insects' has been provisionally accepted for publication in PLOS Computational Biology.

Best regards,

Matthieu Louis

Academic Editor

PLOS Computational Biology

Lyle Graham

Section Editor

PLOS Computational Biology

---

## [Editor Report · Acceptance letter]

PCOMPBIOL-D-24-02038R2

Lateralised memory networks may explain the use of higher-order visual features in navigating insects

Dear Dr Filippi,

I am pleased to inform you that your manuscript has been formally accepted for publication in PLOS Computational Biology. Your manuscript is now with our production department and you will be notified of the publication date in due course.

With kind regards,

Zsofia Freund
